# Do the Current Guidelines for Heart Failure Diagnosis and Treatment Fit with Clinical Complexity?

**DOI:** 10.3390/jcm11030857

**Published:** 2022-02-06

**Authors:** Paolo Severino, Andrea D’Amato, Silvia Prosperi, Alessandra Dei Cas, Anna Vittoria Mattioli, Antonio Cevese, Giuseppina Novo, Maria Prat, Roberto Pedrinelli, Riccardo Raddino, Sabina Gallina, Federico Schena, Corrado Poggesi, Pasquale Pagliaro, Massimo Mancone, Francesco Fedele

**Affiliations:** 1Department of Clinical, Internal, Anesthesiology and Cardiovascular Sciences, Sapienza University of Rome, Viale del Policlinico, 155, 00161 Rome, Italy; andrea.damato@uniroma1.it (A.D.); silvia.prosperi@uniroma1.it (S.P.); massimo.mancone@uniroma1.it (M.M.); francesco.fedele@uniroma1.it (F.F.); 2Department of Medicine and Surgery, University of Parma, 43125 Parma, Italy; alessandra.deicas@unipr.it; 3Surgical, Medical and Dental Department of Morphological Sciences Related to Transplant, Oncology and Regenerative Medicine, University of Modena and Reggio Emilia, 41121 Modena, Italy; annavittoria.mattioli@unimore.it; 4Department of Neuroscience, Biomedicine, and Movement Sciences, University of Verona, Via Casorati 43, 37131 Verona, Italy; antonio.cevese@univr.it (A.C.); federico.schena@univr.it (F.S.); 5Division of Cardiology, University Hospital “P. Giaccone”, University of Palermo, 90133 Palermo, Italy; giuseppina.novo@unipa.it; 6Department of Health Sciences, University of Piemonte Orientale “A. Avogadro”, 28100 Novara, Italy; maria.prat@med.unipmn.it; 7Department of Clinical and Experimental Medicine, University of Pisa, Via Roma, 67, 56126 Pisa, Italy; roberto.pedrinelli@med.unipi.it; 8Section of Cardiovascular Diseases, Department of Medical and Surgical Specialties, Radiological Sciences and Public Health, University of Brescia, 25123 Brescia, Italy; riccardo.raddino@unibs.it; 9Department of Neuroscience, Imaging and Clinical Sciences, Institute of Advanced Biomedical Technologies, “G.D’ Annunzio” University, 66100 Chieti, Italy; sgallina@unich.it; 10Department of Clinical and Experimental Medicine, Division of Physiology, University of Florence, 50121 Florence, Italy; corrado.poggesi@unifi.it; 11Department of Clinical and Biological Sciences, University of Turin, 10043 Turin, Italy; pasquale.pagliaro@unito.it

**Keywords:** heart failure, left ventricular ejection fraction, New York Heart Association classification, acute heart failure, chronic heart failure, phenotypes, pathophysiology, therapy

## Abstract

Heart failure (HF) is a clinical syndrome defined by specific symptoms and signs due to structural and/or functional heart abnormalities, which lead to inadequate cardiac output and/or increased intraventricular filling pressure. Importantly, HF becomes progressively a multisystemic disease. However, in August 2021, the European Society of Cardiology published the new Guidelines for the diagnosis and treatment of acute and chronic HF, according to which the left ventricular ejection fraction (LVEF) continues to represent the pivotal parameter for HF patients’ evaluation, risk stratification and therapeutic management despite its limitations are well known. Indeed, HF has a complex pathophysiology because it first involves the heart, progressively becoming a multisystemic disease, leading to multiorgan failure and death. In these terms, HF is comparable to cancer. As for cancer, surviving, morbidity and hospitalisation are related not only to the primary neoplastic mass but mainly to the metastatic involvement. In HF, multiorgan involvement has a great impact on prognosis, and multiorgan protective therapies are equally important as conventional cardioprotective therapies. In the light of these considerations, a revision of the HF concept is needed, starting from its definition up to its therapy, to overcome the old and simplistic HF perspective.

## 1. Introduction

Heart failure (HF) is a clinical syndrome characterised by specific symptoms and signs due to structural and/or functional heart abnormalities, which lead to insufficient cardiac output and/or increased intraventricular filling pressure [1,2].

Currently, in Europe, HF incidence reaches 5 cases for 1000 people [1,3,4], with a prevalence around 1–2% in adults [1,5,6]. HF is an age-related disease, with an increasing prevalence, ranging from 1%, before the age of 55 up to 10% or more, from the age of 70 years [1,7,8].

In August 2021, the European Society of Cardiology (ESC) published the new guidelines for the diagnosis and treatment of acute and chronic HF [1], according to which the left ventricular ejection fraction (LVEF) continues to represent the pivotal parameter for HF patients’ evaluation, risk stratification and therapeutic management, despite its clear limitations [9,10,11,12,13,14,15,16]. This aspect is due to a vicious circle, nourished by LVEF use, as the main inclusion criterion in the principal HF clinical trials and registries upon which Guidelines are based [1]. LVEF-based HF classification is a simplistic, categorical and cardiocentric approach, which completely neglects HF pathophysiological mechanisms, aetiology and comorbidities. Indeed, regardless of LVEF, HF patients share similar clinical features, with significant overlap among them [9].

Recently, an innovative approach to HF based on phenotypes has been proposed. The use of phenotypes, across the HF spectrum, to categorise HF patients certainly represents a practical approach to bring order to HF complexity [9]. Although a phenotypic approach overcomes the categorical nature of LVEF classification, it may be too simplistic of a setting for a multifaceted and complex disease such as HF. The phenotypic approach considers HF as a continuous and evolving disease, giving more emphasis to new aspects, such as comorbidities, risk factors and disease modifiers. However, it preserves the same cardiocentric view of LVEF: different HF phenotypes are extrapolated by a non-linear relationship between LVEF variability and left ventricle end-diastolic volume. Efforts to optimise HF classification are still needed and more emphasis should be given to multisystemic and progressive multiorgan involvement, which makes HF similar to cancer. Several clinical aspects, such as kidney and lung involvement, anaemia, iron deficiency, liver dysfunction and nervous central system disorders, as well as specific circulating biomarkers are often neglected, although they have a great impact on mortality and morbidity in HF patients, regardless of LVEF [1,17].

## 2. The Pitfalls of Left Ventricular Ejection Fraction

According to the latest ESC Guidelines [1], HF is classified into three types based on LVEF values: HF with preserved EF (HFpEF), if LVEF is ≥50%, in the presence of abnormalities of heart structure and/or function and/or increased natriuretic peptides values, as well symptoms presence; HF with reduced HF (HFrEF), if LVEF ≤40%, and HF with mildly reduced EF (HFmrEF), if LVEF is between 41 and 49% (Table 1). LVEF-based HF classification leads to the diagnosis, stratification and therapeutic management of HF patients [1].

The LVEF-based classification has a lot of limits. Firstly, it does not consider the pathophysiological mechanism and specific aetiology underlying HF [10,11]. Moreover, there are several technical limitations of this parameter. In fact, LVEF is derived from a geometrical assumption, obtained by the Simpson echocardiography 2D technique, which has very high variability, both inter- and intra-observer (for about 13–15%) [10,18]. Moreover, there is also variability among the different imaging techniques used to assess LVEF [19]. LVEF is a load-dependent measure, and it means that the same patient can show different ranges of LVEF, based on its haemodynamic state. At the same time, conditions such as mitral valve insufficiency can overestimate the current LVEF [10,20,21]. For the same reason, a heart suffering from hypertrophic heart disease may show a normal LVEF or a higher LVEF than the normal one. However, if this value is considered alone, it could be misleading as it does not consider the underlying diastolic dysfunction involving myocardium [10,22]. Despite a normal LVEF, this patient has a low cardiac output, and his prognosis is related to several consequences, such as renal impairment and respiratory infections. In fact, the term preserved ejection fraction seems reassuring, but it hides energetic, structural and functional heart abnormalities [23,24]. Of note, LVEF-based classification it is not closely related to HF prognosis, and often, HFpEF patients show worse outcomes than HFrEF’s group, with increased mortality and hospitalisation [10].

From an echocardiographic point of view, LVEF represents only one of the parameters needed to define left ventricular function, but other parameters may be also required, especially now that the HFpEF prevalence is growing [25]. In fact, although HFpEF definition requires the absence of LVEF reduction, different studies show the presence of systolic abnormalities, such as mitral annular plane systolic reduction, mitral annular systolic ejection velocity and longitudinal strain decrease [26,27,28,29]. Furthermore, the alteration of S’, with the tissue Doppler technique, appears closely related to diastolic dysfunction, demonstrating that systolic dysfunction is often present in the HFpEF. Diastolic function is closely linked to the systolic one, and the tissue Doppler study, with the combination of e‘ and S’ evaluation, could be one of the best methods to predict prognosis in patients with HFpEF [30]. Brucks et al. found that diastolic dysfunction was present in >90% of HF patients, regardless of LVEF [9,31]. In fact, both HFrEF and HFpEF patients share a common mechanism: the protein titin’s hypophosphorylation [9,32,33]. Another shared element between HFpEF and HFrEF is the left atrium enlargement. Its function and volume correlate with exercise capacity and predict the outcome regardless of LVEF [9,34,35]. Indeed, HFpEF shows common features of diastolic and systolic dysfunction [36], demonstrating how terms, such as preserved and reduced or systolic and diastolic, may be overlapping and misleading to stratify HF patients.

Several trials evaluated the effect of cardiopoietic stem cell therapy in the treatment of HF [37,38,39,40]. This type of therapy is safe and feasible [37,38], but the explored endpoints, such as adverse events, LVEF improvement, and left ventricular end-systolic and end-diastolic volume were neutral [38]. In this context, a parameter, such as LVEF, may be too rough and operator-dependent to define the ultrastructural modifications that occurred. LVEF does not express myocardial contractility, but it reflects volume modifications, which may be influenced by other factors. Moreover, the follow up time used by trials are too short to observe relevant LVEF modifications. Otherwise, other echocardiographic parameters may be used. For example, global longitudinal strain represents an early parameter used to evaluate the impact of cardiotoxic therapies on myocardial function [41]. Other imaging techniques may be considered to evaluate myocardial response after cardiopoietic stem cell treatment, such as cardiac magnetic resonance, which allows myocardial tissue characterisation.

To overcome the limits and pitfalls of the simplistic and categorical LVEF-based classification, other approaches have been proposed.

The new Universal Definition and Classification of Heart Failure [42] proposes a valid update of the approach to HF patients, considering other aspects beyond LVEF. In fact, Bozkurt et al. identified four HF stages [42]: (i) stage A, including patients likely to develop HF due to risk factors presence, in the absence of cardiac abnormalities and/or signs and symptoms; (ii) stage B, including patients with a pre-HF condition, characterised by structural and/or functional and/or elevated natriuretic peptides or troponins, in the absence of signs and symptoms; (iii) stage C, including patients with current or prior symptoms and/or signs of HF, determined by structural and/or functional cardiac abnormalities; (iv) stage D, or advanced HF, characterised by persistent and refractory symptoms and signs requiring advanced therapeutic approaches, despite optimised medical therapy (Table 1). This staging system considers more elements concerning the overall HF disease than the LVEF-based approach, in particular, signs, symptoms, circulating biomarkers, risk factors and response to therapy.

Another approach based on a new HF perspective has been proposed [9]. HF is defined by different overlapping phenotypes included in a continuous spectrum [9], changing the paradigm from a categoric to a continuous perspective of HF [9] (Table 1). The spectrum takes into consideration other aspects, such as risk factors, trigger factors and comorbidities, being, nevertheless, a cardiocentric approach. In fact, different phenotypes across the HF spectrum are defined by a non-linear relationship between LVEF and left ventricular end-diastolic volume [9]. HF patients’ phenotypes have been successively described to guide therapeutic management [43]. This approach should be pragmatic for the cardiologist, who must follow hundreds of HF patients with different stages of severity and different values of clinical parameters.

The staging systems introduced with the Universal Definition and Classification of Heart Failure [42] and the phenotypes-based HF distinction [9,43] overcome the LVEF, by offering a wider perspective, distinguishing functional, structural cardiac damage, the presence of biohumoral circulating parameters, signs, symptoms, and response to therapies. However, these approaches are still too cardiocentric and superficial. They do not consider HF as a multisystemic disease, giving also little attention to HF pathophysiology.

## 3. The Confusing New York Heart Association Classification

According to symptom onset modalities, two HF types have been identified [1]: (i) acute HF, if symptoms are severe and the patient requires urgent medical evaluation and hospitalisation, and (ii) chronic HF, if HF diagnosis is already known or symptom onset is more gradual. Acute HF often occurs in chronic HF patients who experience an acute decompensation episode, but it may also represent the first manifestation of a new HF onset [1].

The New York Heart Association (NYHA) classification is a functional scale based on HF symptoms. NYHA classification is the oldest classification used for HF, which is based on the evaluation of symptom severity and the ability to sustain effort. It has a prognostic value [44], with an increasing hospitalisation and death risk, according to the rising class number [1,42,45]. Class I includes patients with no symptoms exacerbated by ordinary physical activity. In class II, ordinary activity may cause mild symptoms. Class III is characterised by a significant physical activity limitation, which becomes total in class IV, with patients experiencing dyspnoea at rest (Table 1). NYHA class is simple and user-friendly. It does not require functional and/or structural exams. However, its principal limitation is the large inter-individual variability in the interpretation of symptom severity and related efforts [46], as well as the difficulty to reproduce it because tests do not specify the method of judgment [46]. NYHA classification has low reproducibility and accuracy, and it does not allow a classification based on pathophysiological abnormalities [47]. It is not specific about heart abnormalities because patients with lung alterations or with heart-independent reduced exercise tolerance may also show a high NYHA class. Prognostic stratification and therapeutic management should be driven by an integrated approach of different objective parameters and not by subjective symptoms evaluation: for example, a patient with atrial fibrillation with a normal heart and without multisystemic involvement is more symptomatic than a patient with advanced but compensated HF in optimised medical therapy. However, life expectancy is clearly different beyond symptoms. Despite these evident limitations [47], according to the most recent ESC Guidelines [1], NYHA classification evaluation is pivotal in distinguishing advanced HF and related treatment possibilities [48] from other stages of severity. It guides the implantable cardioverter defibrillator (ICD) and left ventricular assist device (LVAD) implantation, cardiac resynchronisation therapy timing and pharmacological treatment, such as the use of Tafamidis in amyloidotic cardiomyopathy [1].

## 4. Heart Failure Pathophysiology Paradigm: What Is beyond Left Ventricular Ejection Fraction and Symptoms?

HF first involves the heart, but it progressively becomes a multisystemic disease, leading to multiorgan failure and death. In these terms, HF is comparable to cancer: (i) the heart involvement may be considered as the primary neoplastic lesion; (ii) lungs represent the lymph nodes of the heart; (iii) kidneys, liver, bone marrow and nervous central system involvement is comparable to metastasis spread (Table 2). As in cancer, survival, morbidity and hospitalisation are strictly related, not only to the primary neoplastic mass but mainly to metastatic involvement. Moreover, the presence of metastasis hampers therapeutic outlooks, and for this reason, metastasis development prevention is the main strategy for oncologists. As for oncologists, the multiorgan involvement prevention has to be the main strategy to contrast HF progression, also for cardiologists. In HF, multiorgan involvement has a great impact on prognosis; indeed, therapies which slow down this involvement are equally important as conventional cardioprotective therapies.

Given this scenario, LVEF-based classification [1], new classification systems recently proposed [9,42] and NYHA class [44], which are the hinges of the current perspective on HF, are lacking detailed pathophysiological considerations.

From a pathophysiological point of view, other parameters should be considered, beyond LVEF and NYHA classification. Beyond a simplistic categorical classification, there are complex haemodynamic parameters and circulating biomarkers which are common to several HF phenotypes and not always related to LVEF and NYHA classification [9].

HF patients may express a normal or reduced cardiac output [49]. Moreover, most HF patients show a preserved cardiac output, at rest. Regardless of LVEF, cardiac output reserve is often reduced, and patients are unable to improve it during exercise [49,50]. The imbalance of cardiac output reserve has repercussions on ventilation, promotes the renin–angiotensin–aldosterone system (RAAS) and neuro-hormonal hyperactivation, as well as water and sodium retention [49,51]. Reduced cardiac output is a distinctive feature of HFrEF patients. However, HFpEF patients may also show a reduced cardiac output, determined not only by systolic dysfunction but also by impaired ventricular filling and right ventricular overload [49]. The latter one hampers left ventricular filling and cardiac output through the ventricular interdependence principle, nourishing a vicious circle. In this scenario, LVEF evaluation may be significantly misleading because the left ventricular function is deceptively preserved, but left ventricular filling and output are significantly influenced by right ventricular dysfunction [52,53,54].

Pulmonary capillary wedge pressure (PCWP) is a haemodynamic parameter to evaluate left ventricular filling pressure, which is conventionally increased when its value is ≥ 15 mmHg, at rest. Patients with HFpEF often show increased PCWP, which is frequently associated with high central venous pressure (CVP) values [49]. However, patients with advanced HFrEF show increased PCWP and CVP [55,56]. CVP and PCWP are interconnected, and they contribute to lung congestion and pleural effusion, impeding lymph flow [49,53,54]. Moreover, increased CVP pressure has a significant impact on renal venous pressure and glomerular filtration rate (GFR). Renal venous hypertension and consequent trans-renal gradient reduction are associated with venoconstriction and increase in interstitial pressure, contrasting the nephron reabsorption forces [49,57]. Studies using intrarenal Doppler ultrasound [49,58,59] demonstrated a markedly reduced blood flow during diastole when CVP is high. This aspect is associated with GFR decline and diuretic resistance, conditions seen in advanced HF stages and particularly enhanced when renal venous hypertension is associated with reduced cardiac output [49,60].

PCWP evaluation is a precious haemodynamic parameter and an important prognostic predictor. It is useful for the evaluation and management of HF patients [61]. Other parameters of right ventricular dysfunction, such as right atrial/pulmonary capillary wedge pressure (RA/PCWP) ratio, pulmonary artery pulsatility index (PAPI) and right ventricular stroke work index (RVSWI), are associated with poor prognosis in patients with cardiogenic shock and acute HF [62]. In acute HF patients, right ventricular preload haemodynamic parameters impact renal venous pressure and kidney function [60,63]. Recently, life quality and exercise tolerance improvement have been associated with PCWP reduction after interatrial shunt device implantation (IASD), in patients with HF and LVEF ≥40% [64]. Those patients showed a reduction in mortality after IASD implantation [65]. These observations demonstrate the importance of PCWP in HF prognosis.

In this setting, heart catheterisation has very restricted indications in current Guidelines [1], although it is the more reliable test to evaluate PCWP and its variability associated with therapies. It is clearly recommended only in advanced and end-stage HF, when patients are under evaluation for heart transplantation and/or mechanical circulatory support [1]. However, PCWP evaluation may represent an added value in diagnostic assessment, therapeutic management, and prognostic stratification of HF patients. In fact, benefits derived from continuous PCWP monitoring are evident with the CardioMEMS^™^ HF system [66,67]. Other authors [68,69,70] tried to non-invasively estimate PCWP values through speckle tracking echocardiography technique, considering left atrial function and volume measures [68,69], and through cine magnetic resonance imaging [70].

The haemodynamic stress that occurs in HF may also be non-invasively assessed with natriuretic peptides evaluation [71]. The synthesis of natriuretic peptides depends on end-diastolic wall stress, and it is closely related to pressure overload and/or volume expansion. They have a role in the rule in and rule out of patients with suspected HF, in follow-up, risk stratification and therapeutic response of HF patients. However, natriuretic peptides may have clinical use when integrated with other diagnostic tools because they may be influenced by body mass index and renal function. Moreover, natriuretic peptides are elevated also in other conditions, such as ischaemic heart disease and pulmonary embolism, not being a specific HF biomarker [71,72]. Baseline high sensitivity cardiac troponin T and its increase are associated with mortality, disease worsening and acute decompensation, in HF patients [72,73,74,75,76,77,78,79]. Sensitive contemporary cardiac troponin I is also a predictive biomarker of all-cause and cardiovascular mortality, in HF outpatients [73]. Cardiac troponins express myocardial injury. They are valid biomarkers to evaluate response to therapy [71,72] and adverse events prediction [73,76], in patients with HF. Natriuretic peptides and high sensitivity cardiac troponin T values may not always be correlated, demonstrating that they may reflect different pathophysiological pathways, influenced by different factors [72]. For this reason, in HF patients, circulating biomarkers, such as troponin and natriuretic peptides, may be more accurate in diagnosis, prognostic stratification and response to therapy, if included in an integrated system of evaluation. A summary table with clinical, biohumoral, echocardiographic and haemodynamic parameters of HF are listed in Table 3.

## 5. Heart Failure Therapeutic Management: The Misleading Indications

Recently, the 2021 guidelines for the diagnosis and treatment of acute and chronic HF [1] propose four classes of drugs, as first-line HFrEF treatment, to reduce mortality: beta blockers, mineralocorticoid receptor antagonists (MRAs), angiotensin-converting enzyme inhibitor/angiotensin receptor-neprilysin inhibitors (ACEi/ARNIs) and sodium-glucose cotransporter 2 inhibitors (SGLT2i) [1]. LVEF is the main parameter to guide HF treatment. However, the same guidelines emphasise that LVEF is an operator-dependent parameter and includes different LVEF values in the entire spectrum of HF syndrome as a normally distributed variable [1]. The guidelines [1] give large space to HFrEF treatment, while only few indications are given for that of HFpEF and HFmrEF. The only clear therapeutic indication in HFmrEF and HFpEF is the use of diuretics to reduce signs and symptoms of congestion, while further indications are given limited to risk factors control for HFpEF [1]. In fact, the prognosis improvement is observed only in HFrEF patients after conventional HF drug therapy, while important limitations regarding HFmrEF and HFpEF management are evident, in particular regarding the lack of disease-modifying therapies. This aspect cannot be neglected considering three important aspects: (i) HFmrEF/HFpEF represents at least the 40% of HF population [1,80]; (ii) observational studies emphasise that a difference in mortality between HFrEF and HfpEF patients is negligible [1,81]; and (iii) often, patients with HFmrEF develop HFrEF over time [1,82,83]. However, emerging evidence points to the beneficial effect of SGLT2i in reducing the risk for major HF outcomes in patients with HFpEF [84].

Although a multidrug treatment approach is suggested right after HFrEF diagnosis, a relevant problem is forgotten: the treatment adherence. There is clear evidence that adherence to therapy is related to a better prognosis, with a lower number of HF hospitalisations and cardiovascular deaths in chronic HF patients [85,86]. However, the reality is almost different from the theory. In fact, most HF patients have a low therapeutic adherence and not maximally titrated therapy [43,87,88]. This aspect may also have a pathophysiological basis because HF patients often have a complex disease with multiorgan involvement, and, for this reason, they may not be immediately suitable for all HF drugs. Renal involvement, for example, is present in about 50% of the HF population, and it has a determining role in increasing HF-related morbidity and mortality [89]. It is due to a decreased gradient across the glomerular capillary, caused by high CVP [90] and reduced cardiac output. It seems to be more frequently associated with HFpEF than HFrEF, although the negative prognostic impact is greater in the latter group [1,91,92]. This aspect is related to kidney dysfunction and diuretic resistance, which often appear in advanced HF and require increasing doses of loop diuretics and metolazone addition [93]. Kidney dysfunction, in particular acute kidney injury, may represent a limit for a lot of suggested drugs, such as ACEi/ARNIs. Regarding MRAs, in the Eplerenone in Mild Patients Hospitalization and Survival Study in Heart Failure (EMPHASIS-HF), patients treated with Eplerenone showed worsening renal function and an increase in serum potassium level, compared to the placebo group [94]. However, clinical trials showed a benefit for HF patients with CKD in therapy with MRAs [95]. The nephroprotective action has been robustly demonstrated for SGLT2i [96,97,98]. This paradox highlights another limit in the current four-drug therapy approach, recommended by the latest ESC guidelines [1]: treatment is based on LVEF, and it does not consider the different severity stages of HF and kidney involvement. The therapy proposed by the current guidelines is feasible only in a small percentage of patients because it is not always possible to administer all the suggested therapies immediately. As regard, the guidelines [1] do not consider the patient-related haemodynamic state and the high variability rate of renal function, which expose patients to several drug-related collateral effects. Moreover, a real indication regarding the administration and titration timing is missing. Otherwise, a sequential approach may be required, particularly in acute HF. For example, in the case of acute HF, starting with inodilators, such as Levosimendan, it is possible to improve cardiac output and GFR [99,100,101]. Once the patient is stabilised, other drugs may be sequentially added.

The recently proposed phenotypic-based approach [43] suggests that therapeutic management has to be guided by vital parameters, such as systemic blood pressure (BP) and heart rate (HR), in a completely non-specific approach using very tight cut-offs and without clinical contextualisation [43]. For example, high HR should not be always treated as pathological, but it should be investigated in the contest of a reactive response to conditions, such as anaemia and sepsis. It may even be a compensatory and beneficial mechanism, as it occurs in the case of small restrictive hearts, in which cardiac output is HR-dependent. A similar criticism could be addressed to BP therapeutic management: the decision of whether to treat low/high BP or not can be misleading because it is based on a rigid range of values and does not consider the clinical context (i.e., low BP values may be secondary to different conditions, such as hypovolemic state, haemorrhagic shock and sepsis), which may require different specific treatments.

Lastly, the therapeutic decision cannot disregard comorbidities, which play a determining role in the HF prognosis and treatment efficacy. Cachexia and frailty, which are present in almost half of patients with HFrEF and in a consistent number of HFpEF patients, are linked with a worse quality of life and exercise capacity [102]. They also increase the risk of developing HF [1,103]. Sarcopenia is common in chronic diseases, such as cancer or HF, and it is associated with increased mortality and morbidity [1]. Anaemia and iron deficiency are present in 30% to 50% of HF patients, both HFpEF and HFrEF [104], and it may have a large impact on the functional capabilities of HF patients. Studies show that therapies targeting HF-related comorbidities reduce the risk of HF hospitalisation and CV and all-cause death, improving quality of life and symptoms [1,105,106,107].

## 6. Conclusions

An extensive revision of the current HF paradigm is required, from the definition to therapy, due to the limitations exposed (Figure 1). The evaluation of LVEF and symptoms through the NYHA classification is too simplistic and inaccurate for the diagnostic assessment, prognostic stratification, and therapeutic management of HF patients. HF is comparable to cancer because it starts as a heart disease, becoming progressively multisystemic. As for cancer, surviving, morbidity and hospitalisation are related not only to the primary neoplastic mass but mainly to multisystemic metastatic involvement (Table 2). Although several interesting insights were introduced, this new HF perspective may have some limitations: (i) a more comprehensive evaluation of patients requires a complex and multi-speciality assessment, which is not always feasible in terms of time and sources; (ii) the absence of studies to validate this approach in terms of HF prognosis and therapeutic strategy. New classifications [9,42,43] based on a more comprehensive evaluation of HF patients have been proposed, but they remain anchored to a cardiocentric view, neglecting the HF multisystemic nature [108]. Moreover, the current HF point of view disregards the pathophysiological pathways involved in HF progression. Regarding therapeutic management, several critical issues should be mentioned: the absence of disease-modifying therapy for HFmrEF and HFpEF groups, the lack of pragmatism regarding HFrEF treatment, the therapeutic adherence and the lack of attention given to multiorgan involvement therapeutic prevention.

In conclusion, given the HF complexity, a personalised and pragmatic approach to HF patients is required. A transition from a cardiocentric to multisystemic view and from an initial phenotypic classification to a subsequent, well-reasoned, pathophysiological approach may be appropriate to overcome the limitations of current HF perspective. 

## Figures and Tables

**Figure 1 jcm-11-00857-f001:**
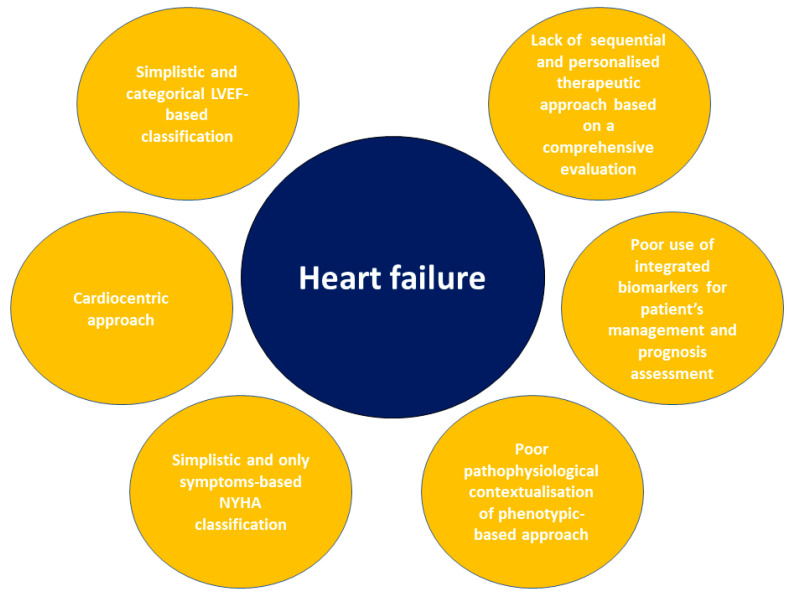
The limitations of the current HF perspective. Summary figure representing the limitations of HF current perspective. LVEF: left ventricular ejection fraction; NYHA: New York Heart Association.

**Table 1 jcm-11-00857-t001:** Table summary showing different HF classifications and staging systems. In the first column, LVEF-based classification by ESC, which differentiates HF into three types, according to ejection fraction: HFpEF, HFmrEF and HFrEF; in the second column, NHYA class, which stages HF according to symptoms severity; in the third column, the evolution stages proposed by the new Universal Definition and Classification of Heart Failure, which considers more parameters, such as risk factors, circulating biomarkers, morphological, functional parameters and response to therapy; in the fourth column, the HF phenotypes classification, which considers several parameters, such as blood pressure and heart rate, in order to obtain a pragmatic treatment strategy.

LVEF	NYHA Class	Evolution Stages	Phenotypes
**(1)** LVEF ≥ 50% (HFpEF) **(2)** LVEF 41–49% (HFmrEF) **(3)** LVEF ≤ 40% (HFrEF)	**(I)** No limitations in normal physical activity **(II)** Mild symptoms in normal activities with slight limitation in physical activity **(III)** Marked symptoms and limitations during daily activities, without symptoms at rest **(IV)** Severe symptoms and limitations, even at rest	**STAGE A** Patients likely to develop HF, due to risk factors presence, in the absence of cardiac abnormalities and/or signs and symptoms **STAGE B (pre-HF condition)** Structural and/or functional and/or elevated natriuretic peptides or troponins, in the absence of signs and symptoms **STAGE C** Current or prior symptoms and/or signs of HF, determined by structural and/or functional cardiac abnormalities **STAGE D (Advanced HF)** Persistent and refractory symptoms and signs, despite OMT, requiring advanced therapeutic approaches	**(1)** Low BP and high HR **(2)** Low BP and low HR **(3)** Normal BP and low HR **(4)** Normal BP and high HR **(5)** AF and normal BP **(6)** AF and low BP **(7)** CKD **(8)** Pre discharge patient **(9)** Hypertensive profile despite OMT

LVEF: left ventricular ejection fraction; HF: heart failure; HFpEF: heart failure with preserved ejection fraction; HFmrEF: heart failure with mildly reduced ejection fraction, HFrEF: heart failure with reduced ejection fraction; NYHA: New York Heart Association; HF: heart failure; OMT: optimised medical therapy; BP: blood pressure; HR: heart rate; AF: atrial fibrillation, CKD: chronic kidney disease.

**Table 2 jcm-11-00857-t002:** Table summary with analogies and differences between cancer and heart failure.

Cancer	Heart Failure
Primary neoplastic mass	Cardiac involvement
Lymph nodes	Lung involvement
Metastasis	Involvement and dysfunction of peripheral organs (i.e., liver, kidneys, brain)
Cancer classification changes slowly	Heart failure is dynamic and can change rapidly over time
Cancer classification is validated regarding therapy and prognosis	Heart failure new paradigm still has not a precise therapeutical and prognostic validation

**Table 3 jcm-11-00857-t003:** Summary table with current major clinical, biohumoral, echocardiographic and haemodynamic parameters useful for heart failure diagnosis and follow-up.

Signs and Symptoms	Circulating Biomarkers	Echocardiographic Parameters	Invasive Haemodynamic Parameters
- Elevated jugular venous pressure - Pulmonary crackles - Pulmonary oedema - Dyspnoea - Orthopnoea - Paroxysmal nocturnal dyspnoea - Reduced exercise tolerance - Fatigue - Increased time to recover after exercise - Hepatomegaly and ascites - Ankle swelling - Breathlessness - Hepatojugular reflux - Third heart sound (gallop rhythm) - - Peripheral oedema	- **BNP**: >80 pg/mL (SR); > 240 pg/mL (AF) - **NT-pro-BNP**: >220 pg/m; (SR); >660 pg/mL (AF)	- **LVEF**: - ≥50% (HFpEF) - 41–49% (HFmrEF) - ≤40% (HFrEF) - **FAC** < 35% - **TAPSE** < 17 mm - **RV S’** < 9.5 cm/s - **e’ septal**: <7 cm/s - **e’ lateral**: <10 cm/s - **Average E/e’**: ≥15 - **TR velocity**: >2.8 m/s - **PASP**: >35 mmHg - **LAVI**: >34 mm/m^2^ - **- LVMI**: ≥149 (M) or 122 (W) g/m^2^ + **RWT** > 0.42	- **LVEDP**: ≥16 mmHg (at rest) - **PCWP**: ≥15 mmHg (at rest) - **PCWP**: ≥25 mmHg (during exercise) - **- CI** < 2.0 L/min/m^2^

BNP: brain natriuretic peptide; NT- pro-BNP: N-terminal pro-hormone of brain natriuretic peptide; SR: sinus rhythm; AF: atrial fibrillation; LVEF: left ventricular ejection fraction; HFpEF: heart failure with preserved ejection fraction; HFmrEF: heart failure with mildly reduced ejection fraction, HFrEF: heart failure with reduced ejection fraction; FAC: fractional area change; TAPSE: tricuspid annular plane systolic excursion; RV S’: systolic velocity of the lateral tricuspid valve annulus; TR: tricuspid regurgitation; PASP: pulmonary artery systolic pressure; LAVI: left atrial volume index; LVMI: left ventricular mass index; RWT: relative wall thickness; LVEDP: Left ventricular end-diastolic pressure; PCWP: pulmonary capillary wedge pressure; CI: cardiac index.

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
