# Peer review of "Do the Current Guidelines for Heart Failure Diagnosis and Treatment Fit with Clinical Complexity?"

_jcm, 2022, doi:10.3390/jcm11030857_

Round 1
Reviewer 1 Report
The authors of this review critically evaluate available diagnostic and prognostic markers in heart failure patients. This is done within the context of the recently published guidlines for HF therapy by the ECS.
Overall, important clinical measures such as NYHA Classification (symptomatic), echorardiografic measures with focus on LVEF and laboratory parameters are critically discussed, also in the context of evaluation of therapeutic efficacy.
The manuscript would need some careful revision by an English native speaker. Also, at times some statements are redundant. For instance, the authors state multiple times throughout the manuscript that LVEF-based classifications are insufficient in characterizing heart failure.
In addition, some references throughout the manuscript cite othe reviews. (e.g. Reference 10, In Section 2 this reference is used multiple times for central statements that are embedded in a commentary of the senior author of this manuscript). I would advise citing the original publications.
The manuscript would also benefit from the inclusion of a table with current clinical measures of heart failure.
Figure 1: The message of figure 1 is not clear and self explanatory. How does the Heart Failure Paradigm relate to the circles that surround it?
Suggestion:
The authors could comment on how clinical studies evaluating new therapies (regenerative therapies/ biologicals) have used LVEF to evaluate efficacy of these therapies and how it impacted the poor outcomes in cell therapy trials.
Author Response
1] The manuscript would need some careful revision by an English native speaker. Also, at times some statements are redundant. For instance, the authors state multiple times throughout the manuscript that LVEF-based classifications are insufficient in characterizing heart failure.
1] We thank the reviewer for the comment. We apologize for the mistakes. We reviewed the text with an English native speaker. We also modified the text, removing some concepts and redundant sentences.
2] In addition, some references throughout the manuscript cite other reviews. (e.g. Reference 10, In Section 2 this reference is used multiple times for central statements that are embedded in a commentary of the senior author of this manuscript). I would advise citing the original publications.
2] We thank the reviewer for the comment. We apologize for the mistake. We reviewed the text and cited original articles.
3] The manuscript would also benefit from the inclusion of a table with current clinical measures of heart failure.
3] We thank the reviewer for the comment. We added a summary table, Table 3, including clinical, biohumoral, echocardiographic and invasive measures of heart failure.
4] Figure 1: The message of figure 1 is not clear and self-explanatory. How does the Heart Failure Paradigm relate to the circles that surround it?
4] We thank the reviewer for the comment. We apologize for the mistake. We reviewed Figure 1, underlying the limitations of current perspective on heart failure.
5] Suggestion: The authors could comment on how clinical studies evaluating new therapies (regenerative therapies/ biologicals) have used LVEF to evaluate efficacy of these therapies and how it impacted the poor outcomes in cell therapy trials.
5] We thank the reviewer for the comment. We added a brief discussion on trials and studies regarding the regenerative/biologicals therapies in HF. We also discuss regarding the possible explication on why LVEF is inadequate to evaluate the efficacy of these therapies.
Reviewer 2 Report
Severino et al analyzed the pitfalls or limits of current LVEF-based classification, NYHA class and the phenotypes-based HF approach, and proposed a new paradigm for heart failure from a multi-systemic view and suggested an extensive revision of heart failure current paradigm from definition to therapy. This manuscript was well-written, and might arouse the interest of a wide range of readers, most importantly, provide reference for the formulation of new guidelines and clinical practice. I have some suggestions.
The authors compared heart failure with tumors, which was the important basis of the new paradigm. The similarities between them deserve further discussion. It is advised to provide a tabular summary. In addition, how the authors view the differences between heart failure and tumors?
In the first few sections of the manuscript, the authors analyzed the limits and pitfalls of LVEF-based classification, new classification systems recently proposed and NYHA class, The text related in the manuscript was lengthy, and the authors should preferably be able to provide a tabular summary of the differences and links between these definitions and the newly proposed one.
Figure 1 depicted the proposed new paradigm for heart failure, and I noticed different colors and circles of different sizes were used in the figure, do these have any deep meaning? This picture seems to be a bit simple. Is there any interconnection between these circles, or a certain part is more weighted? It is recommended that the figure be modified so that it looks more logical and hierarchical.
Compared with the current definitions, the newly proposed paradigm has many advantages, but do the authors believe that there are any shortcomings?
The authors are advised to carefully polish languages and there are some simple grammatical errors in the text, such as ‘lungs may be the lymph nodes and kidneys’ (Page 4 Line 197), ‘by means left atrial function and volume combination’ (Page 6 Line 261), ‘high sensitivity cardiac troponin T as well as its increase are associated with mortality’ (Page 6 Line 272).
Author Response
1] The authors compared heart failure with tumours, which was the important basis of the new paradigm. The similarities between them deserve further discussion. It is advised to provide a tabular summary. In addition, how the authors view the differences between heart failure and tumours?
1] We thank the reviewer for the comment. We have added a tabular summary as suggested (Table 2) with analogies and differences between heart failure and tumours.
2] In the first few sections of the manuscript, the authors analysed the limits and pitfalls of LVEF-based classification, new classification systems recently proposed and NYHA class, The text related in the manuscript was lengthy, and the authors should preferably be able to provide a tabular summary of the differences and links between these definitions and the newly proposed one.
2] We thank the reviewer for the comment. We provided a tabular summary (Table 1) that resumes the main points of the current classification of HF. We also highlighted the differences and possible links among each other.
3] Figure 1 depicted the proposed new paradigm for heart failure, and I noticed different colors and circles of different sizes were used in the figure, do these have any deep meaning? This picture seems to be a bit simple. Is there any interconnection between these circles, or a certain part is more weighted? It is recommended that the figure be modified so that it looks more logical and hierarchical.
3] We thank the reviewer for the comment. We apologize for the mistakes. We modified the Figure 1, in order to resume the main points of the text, reporting the different limitations of current perspective on HF. All the circles have the same colours and dimension, because all the concepts are equally important and relevant.
4] Compared with the current definitions, the newly proposed paradigm has many advantages, but do the authors believe that there are any shortcomings?
4] We thank the reviewer for the comment. We added some sentences in the section Conclusions about possible shortcomings of the proposed HF paradigm.
5] The authors are advised to carefully polish languages and there are some simple grammatical errors in the text, such as ‘lungs may be the lymph nodes and kidneys’ (Page 4 Line 197), ‘by means left atrial function and volume combination’ (Page 6 Line 261), ‘high sensitivity cardiac troponin T as well as its increase are associated with mortality’ (Page 6 Line 272).1)(Abstract): CMR does not need to be spelled out in the abstract (it has come up only once).
5] We thank the reviewer for the comment. We apologize for the mistakes. We revised the text, and we corrected the grammatical errors as suggested.
Round 2
Reviewer 2 Report
This manuscript was largely improved, I have some minor suggestions. The authors are advised to carefully check similar mistakes mentioned below.
- ‘safe e feasible’ in Page 3 Line 124 might be a typo, ‘European Society of Cardiology’ (Page 1 Line 32), ‘global longitudinal strain’ (Page 3 Line 131), ‘cardiac magnetic resonance’ (Page 3 Line 134), ‘magnetic resonance imaging’ (Page 7 Line 286) appeared only once throughout the manuscript and it was not necessary to use their abbreviated form.
- In the fourth column of Table, ‘921) Hypertensive profile despite OMT’ should be revised as ‘9) Hypertensive profile despite OMT’.
Author Response
1) ‘safe e feasible’ in Page 3 Line 124 might be a typo, ‘European Society of Cardiology’ (Page 1 Line 32), ‘global longitudinal strain’ (Page 3 Line 131), ‘cardiac magnetic resonance’ (Page 3 Line 134), ‘magnetic resonance imaging’ (Page 7 Line 286) appeared only once throughout the manuscript and it was not necessary to use their abbreviated form.
1) We thank the reviewer for the comment. We modified the text as suggested. We left the abbreviation ESC for European Society of Cardiology beacause this acronym appears several times in the main text and tables.
2) In the fourth column of Table, ‘921) Hypertensive profile despite OMT’ should be revised as ‘9) Hypertensive profile despite OMT’.
2) We thank the reviewer for the comment. We apologise for the mistake. The Table has been revised.